# Hearing Loss in Stickler Syndrome: An Update

**DOI:** 10.3390/genes13091571

**Published:** 2022-09-01

**Authors:** Frederic R. E. Acke, Els M. R. De Leenheer

**Affiliations:** Department of Otorhinolaryngology, Ghent University Hospital, B-9000 Ghent, Belgium

**Keywords:** Stickler syndrome, hearing loss, hearing impairment, deafness, genetic, *COL2A1*, *COL11A1*, *COL11A2*

## Abstract

Stickler syndrome is a connective tissue disorder characterized by ocular, skeletal, orofacial and auditory manifestations. Its main symptoms are high myopia, retinal detachment, joint hypermobility, early osteoarthritis, cleft palate, midfacial hypoplasia, micrognathia and hearing loss. Large phenotypical variability is apparent and partly explained by the underlying genetic heterogeneity, including collagen genes (*COL2A1*, *COL11A1*, *COL11A2*, *COL9A1*, *COL9A2*, *COL9A3*) and non-collagen genes (*BMP4*, *LRP2*, *LOXL3*). The most frequent type of Stickler syndrome (*COL2A1*) is characterized by a rather mild high-frequency sensorineural hearing loss in about half of the patients. *COL11A1*- and *COL11A2*-related Stickler syndrome results in more frequent hearing loss, being moderate and involving all frequencies. Hearing loss in the rarer types of Stickler syndrome depends on the gene expression in the cochlea, with moderate to severe downsloping hearing loss for Stickler syndrome caused by biallelic type IX collagen gene mutations and none or mild hearing loss for the non-collagen genes. Inherent to the orofacial manifestations, middle ear problems and temporary conductive hearing loss, especially at young age, are also prevalent. Consequently, hearing loss should be actively sought for and adequately treated in Stickler syndrome patients given its high prevalence and the concomitant visual impairment in most patients.

## 1. Introduction

Stickler syndrome is named after Gunnar B. Stickler, a German American pediatrician who worked at the Mayo Clinic in Rochester (USA). In 1965, he described a five-generation pedigree with autosomal dominant inheritance of high myopia, retinal detachment resulting in blindness, hypermobility of the joints, abnormal development of the articular surfaces and premature degenerative changes [1]. Two years later, after a personal communication with a colleague who had seen similar patients, an additional report about the same family was written, adding the presence of mild epiphyseal changes and sensorineural hearing loss [2]. In the subsequent years, several authors published their cases as well and it became clear that hereditary progressive arthro-ophthalmopathy, the initial name of Stickler syndrome, was more prevalent than previously thought. The phenotype of Stickler syndrome includes four body systems symptoms: ocular, skeletal, orofacial and auditory, inherent to the involved collagenous distribution. The main symptoms are displayed in Figure 1.

Hearing loss, the topic of this review, has been variably described in Stickler syndrome and the auditory system is probably the least studied in literature. Audiometry was not performed or reported in numerous publications describing the phenotype of Stickler syndrome patients. Based on a systematic literature review, hearing loss occurred in 63% of the patients [3]. The actual prevalence might be higher than this number as hearing loss was more frequently observed in studies reporting the results of hearing tests compared with subjective perception solely. Consequently, it was not a coincidence that hearing loss was not mentioned in the initial report. However, in patients with visual impairment or at a high risk of becoming visually impaired, the other senses are especially crucial, which stresses the importance of the adequate treatment of hearing impairment. Both conductive hearing loss and sensorineural hearing have been described [4].

Eventually, part of the auditory heterogeneity seen in Stickler syndrome was resolved by the discovery of different genes involved in the syndrome. The first mutation in a Stickler syndrome family was found in the *COL2A1* gene in 1991 [5]. Some years later, disease-causing mutations in the *COL11A1* and *COL11A2* gene could also be associated with a Stickler syndrome phenotype [6,7]. In addition to hearing loss, another clinically distinguishing factor between these three genes is the vitreous appearance, seen on slit lamp biomicroscopy. A membranous type of congenital vitreous anomaly is mainly seen in type 1 Stickler syndrome (STL1, *COL2A1*), whereas a beaded vitreous anomaly is often associated with type 2 Stickler syndrome (STL2, *COL11A1*). In type 3 Stickler syndrome (STL3, *COL11A2*), no ocular symptoms are observed as Col11a2 is not expressed in the eyes [8]. To our experience, skeletal and orofacial symptoms do not differ considerably among the three main types of Stickler syndrome. Additionally, each of the rarer types of Stickler syndrome have their own typical appearance which is summarized in Table 1.

Given the differences in auditory phenotype among the different genes involved in Stickler syndrome, it is important to focus on molecularly confirmed Stickler syndrome patients and distinguish between the different types/genes.

## 2. Hearing Loss in Type 1 Stickler Syndrome (*COL2A1*)

STL1 is caused by a heterozygous mutation in *COL2A1* and confirms the diagnosis in the vast majority of Stickler syndrome cases. A systematic review showed that 52% of STL1 patients had hearing loss, whether based on history taking or audiometry [3], and the majority being sensorineural. This type entails the best prognosis regarding hearing, not only in prevalence but also in severity. It has been reported to be rather mild and mainly affecting the higher frequencies. An age-related typical audiogram of STL1 is provided in Figure 2 and confirms this observation [17]. Progression is not significant when compared to physiological age-related hearing deterioration. Patients will not be referred by newborn hearing screening as the middle frequencies are usually intact, so the timing of onset remains unclear but is supposed to be at childhood age. Consequently, we can state the auditory phenotype of STL1 resembling early-onset presbycusis. Of interest, the amplitude of distortion-product otoacoustic emissions (DPOAE), a test of outer hair cell function, was significantly decreased, more than expected based on the pure tone audiogram [17]. Consequently, (outer) hair cell dysfunction can be observed, which suggests a sensory type of sensorineural hearing loss.

Hearing loss in Stickler syndrome is predominantly sensorineural and symmetric. Conductive and mixed hearing loss is observed in only a minority, most of whom are young children, and is predominantly due to middle ear problems, regardless the type of Stickler syndrome [3,18]. However, especially in patients with a history of cleft palate and associated Eustachian tube dysfunction, these problems can persist into adulthood and necessitate middle ear surgery. Otosclerosis and stapedial fixation have been reported sporadically in STL1 [17,19]. Given the rather high prevalence of otosclerosis, especially in Caucasians, coincidental occurrence of Stickler syndrome and otosclerosis is suspected. In contrast, ankylosis of the footplate and atrophy of the ossicles is a common finding in osteogenesis imperfecta due to a mutation in type I collagen, which is the main collagen in bone [20]. Hypermobility of the tympanic membrane has also been reported in Stickler syndrome, even in different types [4,17,21]. One possible explanation is middle ear problems such as otitis media episodes and transtympanic drain placement in the past. The involvement of type II collagen in the tympanic membrane has also been hypothesized as etiology [4]. The contribution of both has still to be unraveled as not all cases could be linked to middle ear problems [21], and type XI collagen has not been formally confirmed to be present in the tympanic membrane in contrast to type II collagen to our knowledge.

*COL2A1* encodes type II collagen, which is the principal component of articular and hyaline cartilage, and is also present in vitreous and inner ear structures [22]. The early absence of DPOAEs and thus hair cell involvement is supported by the presence of *COL2A1* mRNA in both the inner and outer hair cells, although type II collagen has also been observed in the tectorial membrane of the developing cochlea [23]. STL1 is mainly caused by loss-of-function mutations as the majority of mutations are predicted to result in nonsense-mediated decay [24]. Glycine substitutions, disrupting the triple helical structure of the collagen chain, can also cause STL1. In our experience, the latter are associated with a higher risk of hearing loss.

## 3. Hearing Loss in Type 2–3 Stickler Syndrome (*COL11A1* and *COL11A2*)

Hearing loss in STL2 and STL3 is very similar and will thus be discussed together. Its prevalence is estimated to be between 69% and 83% [3,21]. The average audiogram of hearing-impaired patients exhibiting STL2 or STL3 shows hearing loss that is mainly mild to moderate in the low and middle frequencies, and moderate to severe in the high frequencies. In some patients, the audiogram has a U-shaped configuration due to sparing of the 4–8 kHz frequencies [25,26], which of course might worsen with age. As in STL1, hearing loss has an early onset, but its severity is higher compared to STL1, which makes it more symptomatic at young age. The exact age of onset is still unclear: a significant number of patients passed newborn hearing screening, which is unable to detect mild hearing loss though [18]. Onset or at least progression at childhood age has been reported for both STL2 and STL3 [17,27,28]. No significant progression of the hearing loss could be seen at adult age [21]. Figure 3 shows an age-related typical audiogram for STL2 and STL3 syndrome patients, which is strikingly similar to the audiogram of averaged hearing thresholds for different age groups of STL2 patients reported by Alexander et al. [21]. No temporal bone anomalies could be detected by computed tomography (CT) imaging [29].

A genotype-phenotype correlation can be seen in the *COL11A1* and *COL11A2* mutation spectrum considering hearing. Most heterozygous *COL11A1* mutations linked to Stickler syndrome are supposed to result in a dominant-negative effect [8]. Few heterozygous *COL11A1* mutations resulting in haploinsufficiency have been associated with Stickler syndrome manifestations [30], and the paucity of symptoms in patients with a heterozygous null allele can also be confirmed when looking at families with homozygosity or compound heterozygosity: parental carriers are often asymptomatic or exhibit only mild symptoms [31]. The same pattern can be seen in *COL11A2* mutations. Of interest, biallelic dominant-negative non-glycine *COL11A2* substitutions have been described in non-syndromic hearing loss, being prelingual profound deafness [32]. Profound sensorineural hearing loss has also been observed in the rare autosomal recessive STL2 caused by biallelic *COL11A1* mutations [31].

The genes *COL11A1* and *COL11A2* encode components of type XI collagen, which is widely distributed throughout the body as a regulator of collagen fibrillogenesis in assembly with type II collagen. It is quantitatively present in a rather limited extent though. In addition to its presence in cartilage, essential for skeletal morphogenesis during embryonic development and for articular joint function, it is expressed in other developing structures including the otic vesicle [33]. Col11a1 and Col11a2 demonstrate similar expression patterns except for only marginal detection of Col11a2 in the developing eye [8,34], thus explaining the absence of ocular symptoms in STL3. Several animal models with type XI collagen gene deficiencies have been studied. Based on murine research, it was suggested that mutations in *Col11a1* and *Col11a2* may affect hearing due to impaired structure and function of the basilar and tectorial membrane [35]. Mice with a premature termination codon in both *Col11a2* alleles exhibited a smaller body size, shorter snout and deafness [36]. Mice with a targeted disruption of *Col11a2* demonstrated a frequency-independent moderate cochlear loss and confirmed the involvement of the tectorial membrane by showing impaired collagen fibrillar configuration [37]. Heterozygous haploinsufficiency of *Col11a1* in mice did not result in significant hearing loss [38], as in humans.

## 4. Hearing Loss in Type 4–6 Stickler Syndrome (*COL9A1*, *COL9A2* and *COL9A3*)

Type 4, 5 and 6 Stickler syndromes (STL4, STL5 and STL6) have been attributed to homozygous or compound heterozygous mutations in the type IX collagen genes *COL9A1*, *COL9A2* and *COL9A3*, respectively. Their clinical features have been summarized by Nixon et al. [10]. In summary, STL4 patients showed moderate to high myopia, vitreoretinopathy, sensorineural hearing loss and epiphyseal dysplasia. Of interest, facial symptoms were absent or mild, and none had a palatal defect. STL5 is associated with mild to high myopia, vitreoretinopathy, midfacial hypoplasia, sensorineural hearing loss and few skeletal anomalies. Again, none of the patients exhibited a palatal anomaly. STL6 was associated with moderate to high myopia, variable vitreoretinal symptoms, rather mild midfacial hypoplasia, sensorineural hearing loss, epiphyseal dysplasia and mild intellectual disability. One patient was diagnosed with a high-arched palate, but none had a palatal cleft.

Focusing on the auditory features of these recessive types of Stickler syndrome, sensorineural hearing loss was consistently present in all patients. Moreover, analysis of the audiograms showed striking resemblance: a downsloping curve with mild to moderate hearing loss in the lower frequencies to severe and even profound hearing loss in the higher frequencies. However, variability in severity of hearing loss is apparent. The onset of hearing loss, whether congenital or at childhood age, remains unclear. Both failed and uneventful auditory screening at young age has been reported [12,39], but given the severity of the hearing loss, hearing amplification at a young age is often necessary. Progression of the hearing loss over time has been observed [40,41], and when associating the audiograms of all patients with their age, progression is also suspected, even to profound hearing loss [13,42]. In this way, recessive Stickler syndrome exhibits more frequent and more severe hearing loss than dominant Stickler syndrome.

The involved genes encode the chains for collagen type IX, which is a fibril-associated collagen with interrupted triple helices (FACIT) protein associating with type II and type XI collagen. Murine studies confirmed the involvement of type IX collagen, in addition to type XI collagen, in maintaining the integrity of collagen fibers in the tectorial membrane [43]. Knock-out mice showed progressive hearing loss, already apparent at young age, and morphological changes of the tectorial membrane, starting in the basal turn of the cochlea and progressing towards the apical turn [44]. This is in line with the observed hearing loss in recessive Stickler syndrome: progressive and more pronounced at the higher frequencies.

## 5. Hearing Loss in Stickler Syndrome with Mutations in Non-Collagen Genes (*BMP4*, *LOXL3* and *LRP2*)

Mutations in non-collagen genes are also able to cause a Stickler-like phenotype. It concerns genes with similar expression patterns as the involved collagen genes, especially in the ocular and/or skeletal system, and eventually exhibiting additional symptoms. Based on clinical variability, one might consider a Stickler syndrome phenotype in these patients.

A first example is the *LRP2* gene. A homozygous *LRP2* mutation was found in two siblings with a possible Stickler syndrome diagnosis mainly based on their ocular phenotype, including high myopia, vitreous changes, cataract and esotropia [14]. Mild joint hypermobility was apparent in one patient, in absence of other skeletal symptoms. Additionally, microglobulinuria was detected after molecular diagnosis. Apart from temporary conductive hearing loss based on middle ear problems, no hearing loss was apparent at a young age. *LRP2* mutations are mainly associated with Donnai-Barrow syndrome and facio-oculo-acoustico-renal syndrome. The gene encodes an endocytic transmembrane receptor, which is strongly expressed in the marginal cells of the stria vascularis. Hearing testing in Lrp2-deficient mice showed a progressive hearing impairment [45], which might explain the still normal hearing in the two reported children.

Biallelic *LOXL3* mutations have also been linked to a Stickler syndrome phenotype [15,46]. Several patients have been described, in whom Stickler syndrome was considered as a result of high myopia, hypoplastic vitreous, micrognathia, palatal defects and joint hypermobility. Hearing has not been formally tested in all patients, but no hearing loss was reported apart from in one young patient who experienced conductive hearing loss without further information [15,46]. Of the non-collagen genes, *LOXL3* mutations seem to mimic the classic Stickler syndrome phenotype most, which might be due to the function of the gene. *LOXL3* is a lysyl oxidase gene encoding enzymatic proteins that enable cross-linking of collagen fibers in the extracellular matrix. Tissue-specific ablation of Loxl3 in the inner ear resulted in progressively elevated hearing thresholds for all frequencies [47]. Microscopic analysis revealed hair cell degeneration and secondary spiral ganglion neuron degeneration [47].

A third example is a family exhibiting a heterozygous disease-causing *BMP4* mutation [16]. The *BMP4* phenotype is associated with variable ocular manifestations, ranging from anophthalmia to stigmata associated with Stickler syndrome such as high myopia, retinal detachment and a congenital vitreous anomaly. Retrognathia and a high-arched palate were also seen in this family, as well as renal dysplasia, the latter being typical for *BMP4* mutations but not for Stickler syndrome. Audiograms of the affected members were made available, mainly showing a high-frequency sensorineural hearing loss, more pronounced with older age and more than we would expect from age-related hearing loss alone [16]. In this way, it resembles the hearing loss seen in STL1. *BMP4* encodes a growth factor belonging to the TGF-ß superfamily and proved important in the embryonic development of different organ systems, especially ocular development. In the murine inner ear, Bmp4 heterozygous null mice showed elevated hearing thresholds and circling behavior, suggesting inner ear involvement. Based on microscopic analysis, the number of neuronal processes in the murine organ of Corti as well as the number of vestibular stereocilia was significantly reduced [48].

## 6. Conclusions

Hearing loss is a frequent manifestation of Stickler syndrome. It is often not detected until formally tested as the hearing loss associated with the most prevalent type 1 Stickler syndrome is rather mild and only affects the higher frequencies. It can be distinguished from the hearing loss in type 2 and type 3 Stickler syndrome, which is more pronounced and affects all frequencies. The hearing loss can be explained by the involved types of collagen being expressed in the inner ear, mainly in hair cells and the tectorial membrane, and is thus sensorineural. However, conductive hearing loss has also been reported, mainly linked to the orofacial symptoms including cleft palate and midfacial hypoplasia, affecting Eustachian tube function. Hearing in the rarer types of Stickler syndrome depends on the expression of the involved gene in the cochlea. It has been reported in collagen IX gene mutations, but not consistently in the non-collagen genes causing a Stickler syndrome phenotype. In conclusion, hearing loss should be actively sought for and adequately treated in Stickler syndrome patients given its high prevalence and the concomitant visual impairment in most patients.

## Figures and Tables

**Figure 1 genes-13-01571-f001:**
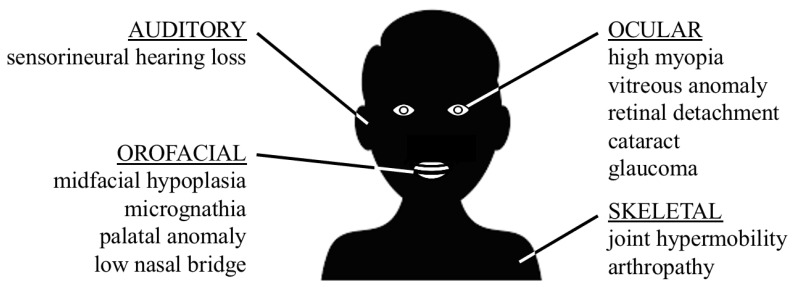
Overview of the four body systems involved in Stickler syndrome, including the main symptoms.

**Figure 2 genes-13-01571-f002:**
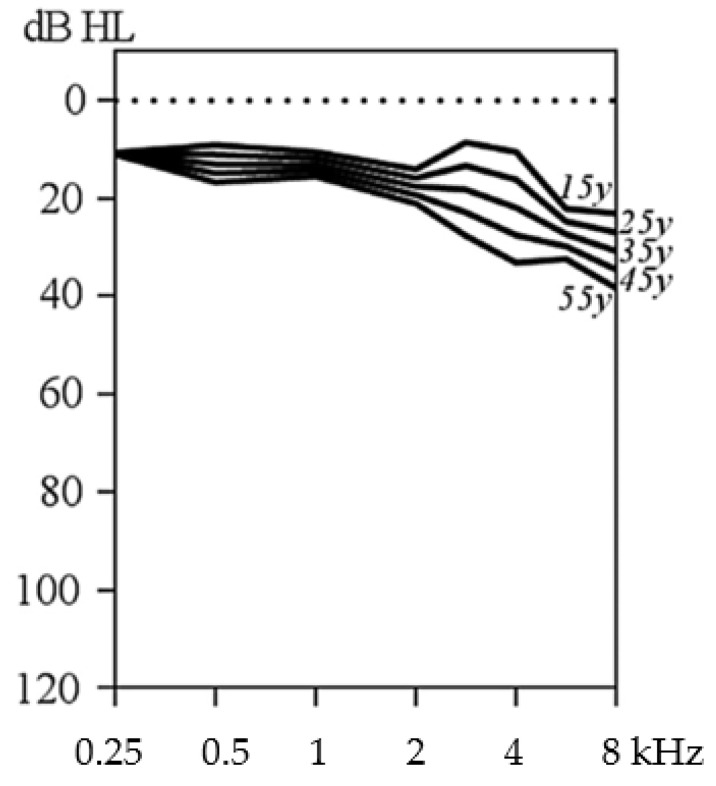
Age-related typical audiogram of patients with type 1 Stickler syndrome, reprinted by permission from Springer Nature: European Archives of Oto-Rhino-Laryngology [17], copyright 2016.

**Figure 3 genes-13-01571-f003:**
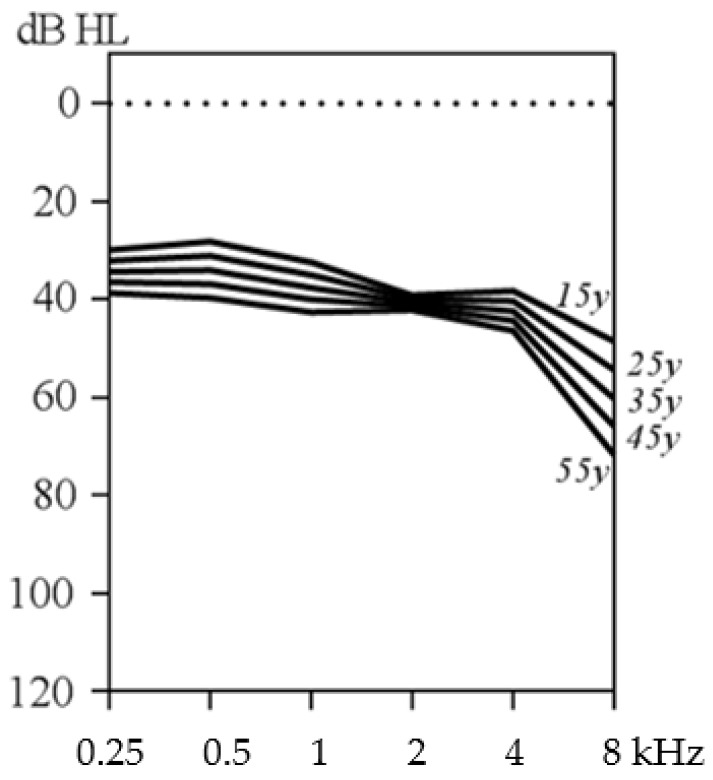
Age-related typical audiogram of patients with type 2 and type 3 Stickler syndrome. The audiogram was created via regression analysis including audiograms from four studies [17,25,26,28] using GraphPad Prism 9 (GraphPad Software Inc., La Jolla, CA, USA).

**Table 1 genes-13-01571-t001:** Overview of the different types/genes of Stickler syndrome with the main distinguishing features (partly based on [9,10], AD = autosomal dominant, AR = autosomal recessive).

Type of Stickler Syndrome	Gene	Inheritance	Main Distinguishing Features	First Description
Type 1	*COL2A1*	AD	Membranous vitreous, Mild hearing loss	[5]
Type 2	*COL11A1*	AD/AR	Beaded vitreous, Moderate (AD) to severe (AR) hearing loss	[6]
Type 3	*COL11A2*	AD	No ocular symptoms Moderate hearing loss	[7]
Type 4	*COL9A1*	AR	Hypoplastic vitreous Moderate to severe hearing loss	[11]
Type 5	*COL9A2*	AR	Hypoplastic vitreous Moderate to severe hearing loss	[12]
Type 6	*COL9A3*	AR	Hypoplastic vitreous Moderate to severe hearing loss	[13]
-	*LRP2*	AR	Microglobulinuria Normal hearing	[14]
-	*LOXL3*	AR	Orofacial defects Normal hearing	[15]
-	*BMP4*	AD	Hypoplastic vitreous Renal dysplasia	[16]

## Data Availability

Not applicable.

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
