# Peer review of "Hearing Loss in Stickler Syndrome: An Update"

_genes, 2022, doi:10.3390/genes13091571_

Round 1
Reviewer 1 Report
In this review, authors presented the the hearing loss issue in the Stickler syndrome. The subject was well presented and discussed with sufficient background and details. Every section was clearly discussed and very informative. For this reason I suggest to publish the manuscript the way it is presented.
Author Response
Dear reviewer, thank you for these favorable comments.
Reviewer 2 Report
35: “can be condensed to” please rather replace by something like “The phenotypes includes four body systems symptoms:” to classify instead of condensing the clinical manifestations of Stickler syndrome
40: “Audiometry was not performed in numerous publications”: not performed or not mentioned ?
45: “However, even mild hearing loss should 45 be actively sought for and addressed in patients at high risk to be or become visually 46 impaired.”
72-74: “Consequently, looking at hearing loss prevalences in the era before molecular confirmation, many are supposed to have type 1 Stickler syndrome.” This is one of the possible explanations. However the prevalence of hearing loss doesn’t directly confirm that most these patients have a COL2A1 pathogenic variant. A suggestion would be to remove this sentence or rephrase it with a more hypothetical meaning.
130: please define “ARTA” in the text
200: Isn’t that “murine” (instead of mural)?
210: as there is literature evidence of pathogenic variants in non-collagen genes please remove “might” here
Figure 1: Could the sheme show the eyes and mouth givent that ocular and orofacial symptoms are listed?
Figure 2: is there any pediatric data that could be used (below 15 yo)
Author Response
Dear reviewer, thank you for your critical reading and constructive comments. We have provided the answers point-by-point below and have modified the manuscript accordingly.
Question 1
Introduction, line 35: “can be condensed to” please rather replace by something like “The phenotype includes four body systems symptoms:” to classify instead of condensing the clinical manifestations of Stickler syndrome
Answer: We have changed “The phenotype of Stickler syndrome can be condensed to four body systems: ocular, skeletal, orofacial and auditory...” into “The phenotype of Stickler syndrome includes four body systems symptoms: ocular, skeletal, orofacial and auditory…”.
Question 2
Introduction, line 40: “Audiometry was not performed in numerous publications”: not performed or not mentioned?
Answer: In fact, both are true. Especially in the earlier publications and the publications focusing on specific symptoms other than hearing loss, the latter was determined by history taking alone. In other publications, mainly in retrospective studies, audiometry was probably performed in a number of patients, but was not reported as such. We have changed “Audiometry was not performed in numerous publications…” into “Audiometry was not performed or reported in numerous publications…”.
Question 3
Introduction, line 45: “However, even mild hearing loss should be actively sought for and addressed in patients at high risk to be or become visually impaired.”
Answer: As there was no question about this phrase, I suppose you would like to rephrase it. We changed it into “However, in patients with visual impairment or at risk to become visually impaired, the other senses are especially crucial, which stresses the importance of adequate treatment of hearing impairment.
Question 4
Type 1 Stickler syndrome, line 72-74: “Consequently, looking at hearing loss prevalences in the era before molecular confirmation, many are supposed to have type 1 Stickler syndrome.” This is one of the possible explanations. However the prevalence of hearing loss doesn’t directly confirm that most these patients have a COL2A1 pathogenic variant. A suggestion would be to remove this sentence or rephrase it with a more hypothetical meaning.
Answer: We agree with this comment and removed the sentence as it is not necessary to understand the paragraph. Moreover, we moved the subsequent sentence “Given the differences in auditory phenotype among the different genes involved in Stickler syndrome, it is important to focus on molecularly confirmed Stickler syndrome patients and distinguish between the different types/genes.” to the end of the introduction.
Question 5
Type 2-3 Stickler syndrome, line 130: please define “ARTA” in the text
Answer: We have removed the abbreviation “ARTA” from the manuscript and replaced it into “age-related typical audiogram” as the audience of Genes might not be familiar with the term ARTA. Moreover, we would also like to announce the addition of the ARTA for type 1 Stickler syndrome, which has already been published before, but for which we received permission to reproduce it. Consequently, an ARTA for type 1 on one side, and type 2 and 3 on the other has been provided in the manuscript.
Question 6
Type 2-3 Stickler syndrome, line 200: Isn’t that “murine” (instead of mural)?
Answer: Thank you. We have replaced “mural” into “murine” throughout the manuscript.
Question 7
Non-collagen genes, line 210: as there is literature evidence of pathogenic variants in non-collagen genes please remove “might” here
Answer: Correct, we changed “Non-collagen genes might also cause a Stickler-like phenotype” into “Mutations in non-collagen genes are also able to cause a Stickler-like phenotype”.
Question 8
Figure 1: Could the scheme show the eyes and mouth given that ocular and orofacial symptoms are listed?
Answer: We have provided eyes and mouth in the picture of Figure 1.
Question 9
Figure 2: is there any pediatric data that could be used (below 15 yo)
Answer: We would like to provide lines for patients <15y or >55y but based on the number of included patients in these age groups and subsequent regression analysis, these lines would not be statistically valid. Consequently, we need to adhere to the textual description, with a variable onset and/or progression at young age, and stabilization at older age.